# FLEXITOKENS: FLEXIBLE TOKENIZATION FOR EVOLVING MULTILINGUAL LANGUAGE MODELS

## ABSTRACT

Language models (LMs) are challenging to adapt to new data distributions by simple finetuning. This is due to the rigidity of their subword tokenizers, which typically remain unchanged during adaptation. This inflexibility often leads to inefficient tokenization, causing overfragmentation of out-of-distribution domains, unseen languages, or scripts. In this work, we develop byte-level LMs with learnable tokenizers to make tokenization adaptive. Our models include a submodule that learns to predict boundaries given the input byte sequence, encoding it into variable-length segments. Existing tokenizer-free methods train this boundary predictor using an auxiliary loss that enforces a fixed compression rate across the training corpus, introducing a new kind of rigidity. We propose FLEXITOKENS, a simplified training objective that enables significantly greater flexibility during adaptation. Evaluating across multiple multilingual benchmarks, morphologically diverse tasks, and domains, we demonstrate that FLEXITOKENS consistently reduces token over-fragmentation and achieves up to 10% improvements on downstream task performance compared to subword and other gradient-based tokenizers.

## 1 INTRODUCTION

Tokenization—the process of segmenting text into discrete units—has been shown to significantly influence language model performance (Ali et al., 2024; Geiping et al., 2024; Land & Bartolo, 2024). Widely used subword tokenization algorithms (Sennrich et al., 2016; Devlin et al., 2019) often overfragment sequences in unseen domains, languages, and scripts. This oversegmentation not only leads to poor downstream performance, but also increased sequence lengths, which contribute to higher computational overhead, memory usage, and inference costs (Ahia et al., 2023; Petrov et al., 2023). In addition, such tokenizers are inherently static and tightly coupled with the language model; they do not adapt when the language model is fine-tuned, e.g., fine-tuning Llama 2 models is subpar for coding tasks (Dagan et al., 2024; Minixhofer et al., 2024), and unseen scripts (Li et al., 2023). Eliminating the reliance on static subword tokenizers has, thus, gained momentum in recent literature by directly modeling bytes (Xue et al., 2022; Al-Rfou et al., 2018; Wang et al., 2024), although this comes with the overhead of longer sequence length.

To address the increase in sequence length in byte-level language models, various papers introduce a tokenization module within the LM to segment bytes into patches (?Nawrot et al., 2022; Ahia et al., 2024; Pagnoni et al., 2024; Nawrot et al., 2023; YU et al., 2023). As opposed to subword tokenizers, this module is typically learned via gradients alongside the LM with an auxiliary loss to achieve a desired *compression rate* of the input sequence during training. This compression rate, while controllable, is predetermined and fixed during pretraining, which again hampers adaptation to new distributions (see Figure 1). For example, an LM trained with a fixed compression rate on a general domain may over-tokenize samples in specialized domains like Medicine or morphologically rich languages like Turkish that contain longer words. Conversely, it may undertokenize samples in programming languages or logographic languages like Chinese, where distinct semantic units may be inappropriately merged.

To enable flexible adaptation of gradient-based tokenizers, we propose a new training objective, which *relaxes the need to have a fixed* compression rate. Instead, we define a lower bound on the compression rate that every input sequence should have and introduce a hinge-like loss to optimize the tokenizer with this rate. By not penalizing the tokenizer when the compression rate is higher than

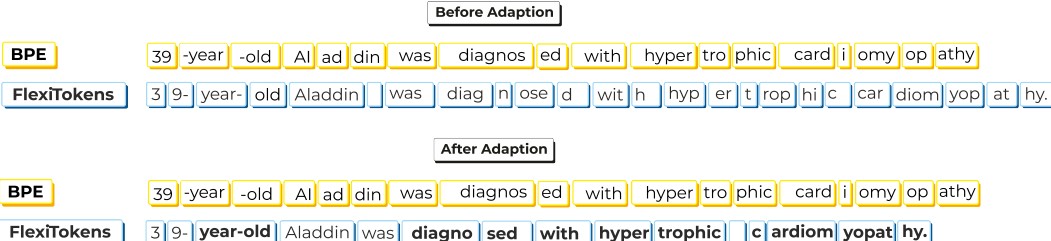

Figure 1: We present an example of tokenized medical text, where FLEXITOKENS produces a less fragmented sequence of tokens than BPE. Unlike BPE, which applies a fixed tokenization, FLEXITOKENS adapts its tokenization to the medical domain, capturing domain-specific patterns more effectively.

this rate, our method allows for the segmentation to be flexible to the input sequence. When the LM is fine-tuned, this loss allows the tokenization to effectively adjust to the target distribution without leading to overfragmentation. We call our method FLEXITOKENS.

We evaluate our proposed approach on multiple multilingual benchmarks and morphologically diverse tasks (§4). FLEXITOKENS consistently shows superior performance compared to baselines while improving average compression rate, thereby improving inference runtime. We also show that while maintaining a fair fragmentation rate across all our pretraining languages, FLEXITOKENS can be easily adapted to unseen languages and scripts without leading to overfragmentation. Our analysis shows that our method often updates the tokenizer to recover semantically meaningful tokens relevant to the task or domain after adaptation, whereas the baselines, being not updatable, overtokenize.

## 2 FLEXITOKENS

We build a byte-level LM with a learnable tokenization module integrated within the model. FLEXI-TOKENS allows the model to adjust its learned tokenization strategy to the structure and distribution of the task and input data. Our model builds on a hierarchical transformer backbone (Nawrot et al., 2022), which is generally used to efficiently handle long sequences in tokenizer-free models (Nawrot et al., 2022; 2023; Ahia et al., 2024; Pagnoni et al., 2024; Hwang et al., 2025). Despite being learnable, the resulting tokenization modules in prior work remain bound to the decisions made during pretraining, even when the model is trained or finetuned further. This inherently limits their ability to adapt to new domains, languages, or evolving data distributions, where the originally learned segmentation might no longer be optimal.[1] Below, we first describe the key components of our model adopted from prior works on tokenizer-free models (§2.1) and then introduce the modifications we make to enable dynamic and equitable tokenization (§2.2).

### 2.1 MODEL ARCHITECTURE AND OVERVIEW

The hierarchical transformer architecture (Figure 5) was designed to handle long sequences in byte-level language models. It consists of a tokenization or downsampling layer that takes byte sequences as input and chunks them into tokens or *patches*. It is then followed by a standard language modeling module, which in turn is followed by an upsampling layer to predict bytes as outputs.

The **tokenization submodule** processes input byte sequences using a lightweight transformer (1 to 2 layers) that maps each byte in an input byte sequence $x_1, \ldots, x_N$ to hidden states. Taking these hidden states as input, a boundary predictor then estimates the probability $\hat{b}_t \in [0, 1]$ of predicting a segment boundary at each position $t$. It is implemented using an MLP followed by a sigmoid function. To obtain discrete boundary decisions $b_t \in \{0, 1\}$ while preserving differentiability, a hard Gumbel sigmoid re-parameterization of the Bernoulli distribution is employed. Since this module is differentiable, the segmentations are learned along with the rest of the model.

---

[1]This issue is also present in subword tokenizers like BPE. Prior work typically handles this issue with heuristics like retraining and replacing the entire tokenizer during adaptation (Minixhofer et al., 2024).

Given the predicted boundaries, the **language modeling module** pools the byte-level hidden states between segment boundaries to construct a sequence of token-level representations. These representations are then passed through a stack of transformer layers to obtain another sequence of hidden representations. In a typical LM, this module is equivalent to the transformer blocks without the input and output embedding layers.

Finally, the **upsampling module** converts the outputs from the middle LM block to byte-level probabilities. The token-level representations from the middle block are first upsampled to match the original input resolution via duplication and added to the initial byte-level representations (from the tokenization module) using skip connections. These representations are then passed through another lightweight transformer (1 to 2 layers), an unembedding layer, and a softmax to produce a probability distribution over the byte vocabulary (of size 256) at each step. The LM loss is computed over the byte vocabulary using these distributions. This modeling backbone is based on the one described in Nawrot et al. (2023); we refer the reader to this paper for a more detailed description.

To prevent the boundary predictor from collapsing and trivially predicting each position $t$ as a boundary, prior work (Nawrot et al., 2023; Ahia et al., 2024) added a regularizer to the LM objective: $-\log \mathrm{Binomial}(\alpha; N, k)$ where,

$$\mathrm{Binomial}(\alpha; N, k) = \binom{N}{k} \alpha^k (1 - \alpha)^{N-k}, \quad \text{and} \quad k = \sum_N b_t. \tag{1}$$

$\alpha \in [0, 1]$ is a hyperparameter that controls the expected boundary rate. This loss is lowest when $k$ is close to $\alpha N$, which is the mode of the Binomial distribution. In other words, $\alpha$ controls the compression rate of the input sequence to approximately $\frac{1}{\alpha} \times$. Setting $\alpha = 0$ will cause no boundaries to be predicted, and with $\alpha = 1$, the model learns to predict every position to be a boundary. More recent work has also employed similar objectives to enforce a compression rate (Hwang et al., 2025). This loss is added to a cross-entropy loss for next-byte prediction to train the model and tokenizer in an end-to-end fashion.

## 2.2 FLEXITOKENS

In contrast with subword-based models like BPE, LMs with gradient-based tokenization can learn to segment input text in a way that best represents the underlying data distribution for language modeling. Furthermore, prior work has shown that it allows better controllability over segmentation rates over different languages when training multilingual models by simply employing different boundary predictors with different compression rates per language or script (Ahia et al., 2024), leading to more equitable tokenization (Petrov et al., 2023). However, even within a language, different subsets, such as different domains, might require different compression rates to optimally encode the input. But the expected compression rate is predetermined by the hyperparameter $\alpha$ with little room for variation. Furthermore, when adapting the LM to new distributions, such as a new domain or a new language, bound by the loss in Equation 1, the compression rate does not update to the requirements of the target distribution.

The ideal solution to address this issue is to get rid of the hyperparameter $\alpha$ (and the binomial loss) and simply minimize the predicted number of boundaries per byte, that is, $\frac{k}{N}$. If optimized well, this loss will find the right balance between compression and minimizing the LM loss. However, in our early experiments, we observe that this loss quickly decreases to $0$, predicting no boundaries. To prevent this behavior, we modify this loss to

$$\mathcal{L}_{\mathcal{BP}} = \max\left(\frac{k}{N} - \alpha, 0\right) + \max\left(\beta - \frac{k}{N}, 0\right), \text{ where } \beta = \alpha - \lambda\sigma \tag{2}$$

$\sigma$ represents the standard deviation of tokenization rates over multiple samples in a given language. $\lambda$ is a hyperparameter. This loss introduces a constraint on the boundary rate at $\beta \leq \frac{k}{N} \leq \alpha$. If the boundary rate falls between $\beta$ and $\alpha$, this loss will become $0$, reducing further incentive to compress by not penalizing the model. In contrast, the binomial loss forces the rate to be extremely close to $\alpha$, penalizing both an increase and a decrease at all times. Indeed, we observe in our experiments that there is higher variance in the segmentation rates of different samples. Furthermore, during finetuning, we observe changes in the compression rates, showing that the tokenization indeed adapts

to the task. We refer to the flexible tokens learned through our proposed loss and the resulting model that predicts flexible tokens as FLEXITOKENS.[2]

To encode the same information, different languages require different number of bytes, where non-Latin languages (e.g., Indian languages) may require up to 4 bytes per character. When training multilingual models, setting one $\alpha$ for all languages will lead to text in some languages getting segmented into much longer sequences. To alleviate this issue, Ahia et al. (2024) proposed adding a different boundary predictor per language with its own $\alpha$ defined to make the compression rates uniform across languages. A unique boundary predictor per language, however, requires determining or predicting the input language to route the input to the appropriate predictor. It also makes it challenging when the input text contains multiple languages (in case of code-mixed text). Our experiments reveal that training one shared boundary predictor with a different hyperparameter $\alpha_L$ for each language $L$ leads to the same performance. Hence, we train a multilingual model with the following training objective.

$$\mathcal{L} = \sum_{i=1}^{N} -\log p_\theta(x_i \mid x_{<i}) - \sum_{\mathcal{M}} \mathbb{I}(\text{language}(\mathbf{x}) = L)\mathcal{L}_{BP_L} \tag{3}$$

where $\mathcal{M}$ is the set of all languages in the training set.

**Determining $\alpha_L$ and $\beta_L$:** We define an anchor language A[3] and set $\alpha_A$ as a hyperparameter. We assume access to a small $n$-way parallel corpus[4] between $A$ and every other language $L$ in our training set.[5] We compute the mean sequence length (in bytes) $\mu_A$, $\mu_L$, and standard deviation $\sigma_A, \sigma_L$ over this dataset. We set $\alpha_L$ to be $\alpha_A \frac{\mu_A}{\mu_L}$, and define the lower bound $\beta_L$ as $\alpha_L - \lambda\sigma_L$. Intuitively, if $L$ uses more bytes to represent the same information as $A$, its compression rate should be higher (and hence $\alpha_L$ lower). In summary, only $\alpha_A$ and $\lambda$ are hyperparameters; the others are derived from them.

## 3 EXPERIMENTAL SETUP

### 3.1 DATASETS

We validate our proposed approach in a multilingual setting. We train models with four scripts and six languages: Latin script (English and Spanish), Cyrillic (Russian and Ukrainian), Devanagari (Hindi), and Telugu script (Telugu). These scripts cover a diverse range of typologies and byte complexities. For example, Latin script needs 1 byte per character in Unicode, whereas Russian and Telugu characters need up to 2 and 3 bytes, respectively. To make tokenization rates similar across all languages, all these languages require different amounts of compression.

For pretraining, we sample the first 2.06M documents from FineWeb (Penedo et al., 2024a) for English and Spanish, using the first 10K documents as the validation set. For all other languages, we sample the first 1.65M documents from FineWeb 2 (Penedo et al., 2024b), again using the first 10K documents for validation. A breakdown of the training set sizes is shown in Figure 6 (in §B).

For downstream evaluations, we finetune on the following generative and understanding tasks: (1) *OPUS-100* (Tiedemann, 2012): machine translation, (2) *XNLI* (Conneau et al., 2018): natural language inference, (3) *SIB-200* (Adelani et al., 2023): topic classification, (4) *Multilingual Sentiment* (clapAI, 2024): multi-domain sentiment analysis, (5) *WikiANN* (Pan et al., 2017): named entity recognition, (6) *Indo-Aryan Language Identification (ILI)*[6] (Zampieri et al., 2018): dialect classification, (7) *SwitchLingua* (CS) (Zampieri et al., 2018): code-switched text classification; (8) *Medical Abstracts Text Classification* (Schopf et al., 2022) and (9) *Irony detection* in Tweets containing emojis (Rohanian et al., 2018). We provide more details on each dataset in §B. We also considered evaluating our models on common tasks like MMLU (Singh et al., 2024) and INCLUDE (Romanou et al., 2024) however, we got random scores due to the small size of our models (See Table 11 in Appendix C).

---

[2]We use the term interchangeably to refer to our model and proposed loss.

[3]We choose A as English in all our experiments. This choice is arbitrary; choosing another language will change the $\beta$ values but will not influence the final results.

[4]This computation can also be done with a pairwise parallel dataset with the anchor language with slight modifications.

[5]This parallel dataset is not used for training the model.

[6]https://github.com/kmi-linguistics/vardial2018

### 3.2 HYPERPARAMETERS

To understand the impact of sequence compression on model's performance, we explore multiple compression rate configurations. Our main results use $3\times$ compression rate for our anchor language, English (i.e. $\alpha = 1/3$). We also compare with $5\times$ and $10\times$. The corresponding values of $\alpha_L$ and $\sigma_L$ for all languages are in Table 1. We compute $\beta_L$ using the FLORES-200 dataset (Costa-Jussà et al., 2022), which contains parallel sentences in 200 languages. We empirically set $\lambda = 3$; we show comparisons with other values in §4. In our experiment with adapting our model to an unseen script (for Urdu), we set $\beta$ to have the same value as Telugu, which has the highest compression rate of all the languages we experimented on, assuming no available training dataset in the unseen language.

**Model Architecture and Pretraining**  We pretrain a model with 119M parameters. We follow Ahia et al. (2024) to create a 16-layer hierarchical transformer. The tokenization and upsampling submodules each consist of 2 transformer layers, while the language modeling submodule contains 12 transformer layers. The input embedding dimension is 768. All transformer layers have a hidden size of 768, with a feed-forward intermediate dimension of 3072, and we use 12 attention heads in the self-attention mechanism. All other parameters follow Ahia et al. (2024), except for the boundary predictor: instead of multiple predictors, we use a single 2-layer MLP as the boundary predictor. See Figure 5 for an overview of our model's design. We use the hierarchical transformer implementation by Nawrot et al. (2022), since their training code is publicly available and it has been successfully adopted by other works (Fleshman & Van Durme, 2023; Ahia et al., 2024). An alternative implementation, H-Net (Hwang et al., 2025), does not provide its training code, and has also been found to be unstable during reproduction by previous studies (Main, 2025). We note that our method/loss for flexible tokenization is generalizable to any hierarchical transformer model with a tokenization submodule.

During pretraining, we use a chunk size of 512 bytes. We train for 100K steps with a cumulative batch size of 512 across 2 H100 GPUs with 9000 warmup steps. Optimization is performed with Adam (Kingma & Ba, 2014), a cosine learning rate scheduler (with maximum learning rate of 5e-5), and gradient clipping set to 0.25. See §B for a full list of hyperparameters.

Table 1: $\alpha_L$ and $\sigma_L$ values for each language in our training dataset, computed using FLORES-200. The upper bound $\beta_L$ in Equation 3 is computed as $\alpha_L - \lambda\sigma_L$)

| Configuration | en | es | ru | uk | hi | te |
|---|---|---|---|---|---|---|
| FLEXITOKENS $10\times$ | 0.1 / 10 | 0.08 / 12.12 | 0.05 / 19.92 | 0.053 / 18.70 | 0.039 / 25.62 | 0.037 / 26.91 |
| FLEXITOKENS $5\times$ | 0.2 / 5 | 0.17 / 6.06 | 0.1 / 9.96 | 0.107 / 9.35 | 0.078 / 12.81 | 0.074 / 13.45 |
| FLEXITOKENS $3\times$ | 0.333 / 3 | 0.28 / 3.64 | 0.167 / 5.98 | 0.178 / 5.61 | 0.13 / 7.68 | 0.124 / 8.07 |
| $\sigma$ | 0.023 | 0.019 | 0.011 | 0.012 | 0.009 | 0.008 |

### 3.3 BASELINES

We compare with two baselines: (1) a model trained with a BPE tokenizer and (2) a byte-level model whose boundary predictor is trained with a binomial loss as described in Nawrot et al. (2023) (BINOMIAL). For fair comparison with the BPE model, we match its overall parameter size with FLEXITOKENS. We train a BPE tokenizer with a vocabulary size of 50K on the same amount of dataset from each language; it achieves a compression rate of $4.4\times$ on English.[7] To match total parameters (embeddings + transformer layers), we train the LM with 5 Transformer layers.[8]

---

[7]Note that BPE models cannot be controlled to have desired compression rates across all languages due to their inherent frequency-based training process (Ahia et al., 2023).

[8]We conducted early experiments with training BPE-based models by matching English's compression rate to $3\times$ compression rate, but they resulted in vocabulary sizes of 10K, which performed poorly.

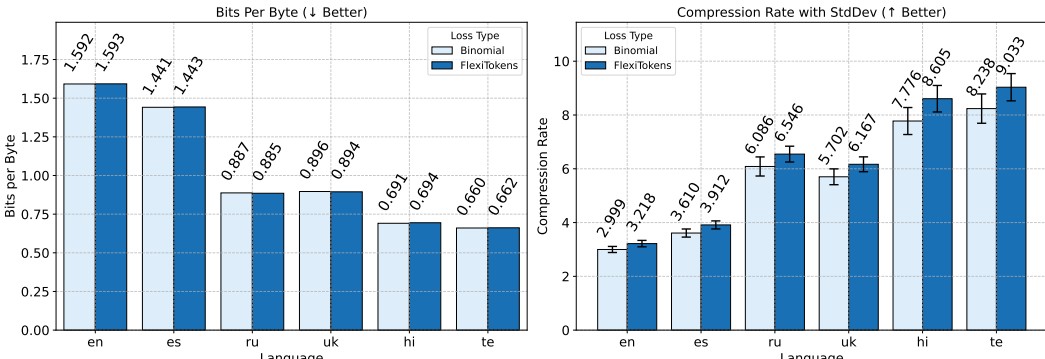

Figure 2: FineWeb Test BPB (↓), Compression rate (↑) and Compression variance (↑) of FLEXITO-KENS compared to the BINOMIAL variant with $\alpha_A = 0.3$ and $\lambda = 3$. Higher compression rates result in fewer tokens, which in turn leads to a more efficient model. FLEXITOKENS matches BINOMIAL on BPB while outperforming considerably on compression.

## 4 RESULTS AND ANALYSES

We evaluate our pretrained model using bits per byte (BPB) (Graves, 2013) and the finetuned models using task-specific metrics, mostly accuracy and F1-score. We provide a summary of the results for the pretrained models in Figure 2 and Figure 3, and for the finetuned models in Table 3, Table 2, and Figure 4, with details in Appendix C.

**Pretraining with FLEXITOKENS leads to better compression**   As shown in Figure 2, our method maintains the same BPB performance as BINOMIAL on the FineWeb test sets while achieving a substantially higher average compression rate, which in turn increases inference speed by requiring fewer tokens. From our experiments, we do not observe any strong correlation between BPB and compression rate.

We also observe a higher variance in compression rates of FLEXITOKENS, implying higher flexibility in how input sequences are fragmented. This variation—which is much lower in baseline models—alongside the higher compression rate on average underscores FLEXITOKENS' ability to dynamically adapt its tokenization patterns to its input. In Figure 3, we compare average number of tokens required to represent the same information in different languages by different tokenization methods. Our method remains as equitable as BINOMIAL using a similar number of tokens for all languages. In comparison, BPE shows high variability with included languages like Hindi and Telugu requiring twice as many tokens. An unseen language (Urdu) requires 6 times as much.

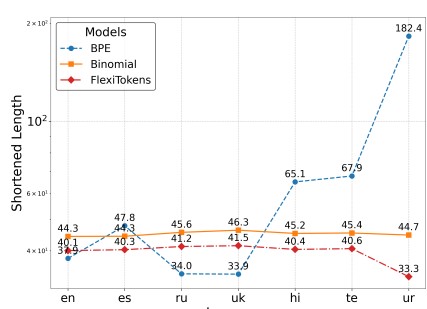

Figure 3: Average number of tokens per sample obtained in the FLORES dataset with different tokenization algorithms. FLEXITOKENS consistently produces the least number of tokens while maintaining balance across languages, even for the unseen language Urdu. BPE over-fragments seen (Hindi, Telugu) as well as unseen languages (Urdu).

**FLEXITOKENS adapts tokenization and boosts performance across tasks and domains.**   In Tables 3 and 2, we report task-specific metrics after finetuning our pretrained models on several downstream tasks across different domains and the corresponding compression rates per language and task in Figure 4. FLEXITOKENS outperforms all baselines on the majority of tasks, including the BPE baseline with a much higher compression rate. Our method obtains performance improvements of over 3 points on SIB-200 (Table 8 in Appendix C) and XNLI compared to BINOMIAL while improving compression across all tasks. Notably, we observe up to 5 points improvement over BPE on our

Table 3: WikiANN (NER), XNLI and other tasks' F1 Score and Accuracy and for $3\times$ Compression Rate. FLEXITOKENS outperforms all baselines on XNLI and NER respectively. Notably, it achieves approximately a 3-point gain on XNLI for Urdu——an unseen language script——compared to BPE

| **NER F1 Score** | | | | | | | |
|---|---|---|---|---|---|---|---|
| **Model** | **en** | **es** | **ru** | **uk** | **hi** | **te** | **Avg** |
| BPE | 52.30 | 67.70 | 64.94 | 74.99 | 60.23 | 48.18 | 61.39 |
| BINOMIAL | 63.80 | 75.06 | 67.59 | **78.06** | 61.21 | 48.31 | 65.67 |
| FLEXITOKENS $\lambda1$ | 63.07 | 76.12 | **68.30** | 77.94 | **62.26** | **51.74** | **66.57** |
| FLEXITOKENS $\lambda2$ | **63.96** | **76.23** | 67.55 | 77.99 | 62.24 | 48.13 | 66.02 |
| FLEXITOKENS $\lambda3$ | 63.73 | 75.45 | 68.25 | 78.01 | 61.97 | 50.88 | 66.38 |

| **XNLI Accuracy** | | | | | | | |
|---|---|---|---|---|---|---|---|
| **Model** | **en** | **es** | **ru** | **hi** | **te** | **ur** (OOD) | **Avg** |
| BPE | 73.09 | 69.9 | 65.95 | 61.48 | **68.00** | 54.11 | 65.42 |
| BINOMIAL | 72.87 | 70.28 | 65.93 | 62.26 | 66.11 | 54.79 | 65.37 |
| FLEXITOKENS $\lambda1$ | **73.51** | 70.22 | 66.47 | **62.42** | 67.11 | 56.99 | 66.12 |
| FLEXITOKENS $\lambda2$ | 73.21 | **70.84** | **66.97** | 62.16 | 66.71 | **57.58** | **66.25** |
| FLEXITOKENS $\lambda3$ | 73.35 | 70.22 | 66.75 | 62.36 | 67.82 | 57.33 | 66.31 |

| **Med. Abs./Irony/CS/CS/ILI - Accuracy** | | | | | | | |
|---|---|---|---|---|---|---|---|
| **Model** | **Med. Abs.** **en** | **Irony** **en** | **CS** **en-es** | **CS** **en-hi** | **ILI** **hi** | **-** **-** | **Avg** |
| BPE | 57.68 | 67.86 | **92.48** | **87.36** | 89.06 | - | 78.89 |
| BINOMIAL | 62.81 | 67.60 | 91.62 | 84.98 | 89.47 | - | 79.30 |
| FLEXITOKENS $\lambda1$ | 62.92 | 68.37 | - | - | 89.58 | - | 73.62 |
| FLEXITOKENS $\lambda2$ | 62.74 | 68.75 | 91.37 | 86.53 | **90.33** | - | 79.94 |
| FLEXITOKENS $\lambda3$ | **63.19** | **69.26** | 92.11 | 86.41 | 89.55 | - | **80.10** |

generative task (translation). We also notice that as we increase $\lambda$, performance tends to also increase. This is because a higher $\lambda$ allows a wider margin for the model to find the optimal compression rate.

Table 2: COMET scores on OPUS-100 machine translation task. FLEXITOKENS outperforms across all languages.

| Model | en-es | en-ru | en-hi |
|---|---|---|---|
| BPE | 59.46 | 52.53 | 50.94 |
| BINOMIAL | 63.05 | 57.33 | 54.35 |
| FLEXITOKENS $\lambda3$ | **64.08** | **57.76** | **54.73** |

Analyzing compression rates across tasks and languages in Figure 4, we observe that BINOMIAL maintains rates closer to the initial $\alpha$, but this effect diminishes for non-Latin languages such as Hindi and Telugu, which are structurally distant from Latin scripts. These languages show both higher average compression and greater variance with FLEXITOKENS.

Qualitative analysis reveals consistent tokenization patterns across topic classification tasks like SIB-200 and Medical Abstracts, where compression remains stable across examples. In contrast, tasks such as XNLI exhibit compression spikes across all languages, indicating that some tasks benefit from more compression than others. In the Irony Classification task, FLEXITOKENS effectively tokenizes emojis with higher compression, preserving their semantic meaning. Following adaptation to the medical domain (Figure 1), we also find that medical terms are tokenized in unison as whole words, reducing fragmentation and better aligning with expected domain-specific vocabulary.

**Adaptive tokenization to unseen scripts boosts performance without overfragmentation** In Table 3, we extend our evaluation to Urdu, a low-resource Indo-Aryan language that shares linguistic commonalities with Hindi but uses a different script, not included in our pretraining dataset. We see

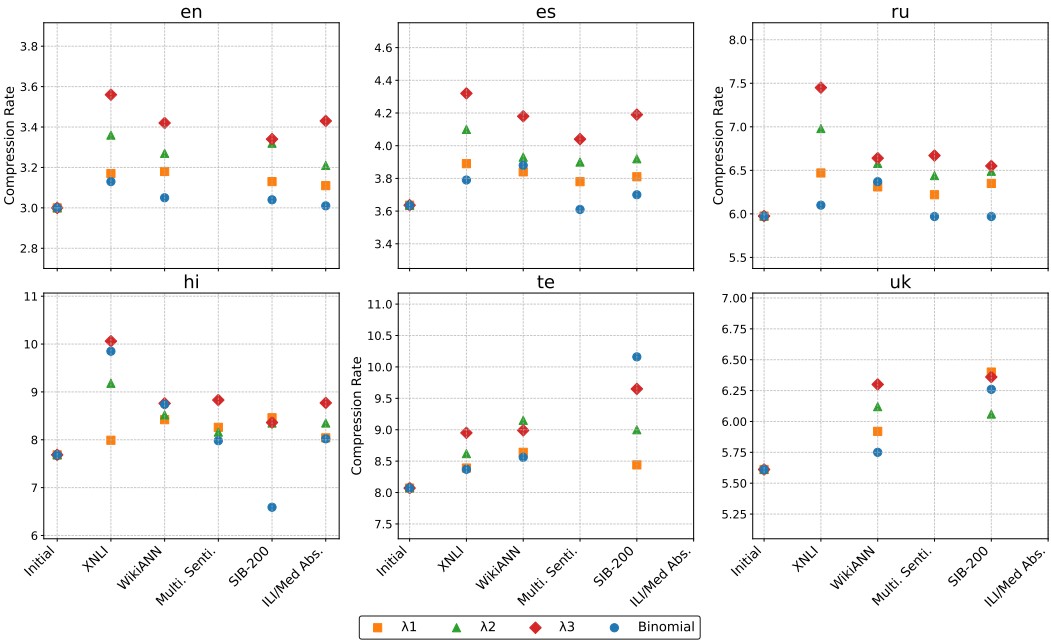

Figure 4: Compression rate changes with FLEXITOKENS across multiple tasks. *Initial* is the base compression rate before pretraining. Compression rate for BINOMIAL remains relatively low while we also see a spike for task like XNLI

that FLEXITOKENS outperforms BPE with more than 3 points after finetuning. Qualitative evaluation on the XNLI inputs (Table 17) reveals that our approach finds more compressed and semantically meaningful tokens compared to baselines (numbers and words). BPE tokenizer tokenizes Urdu with more 6× tokens than FLEXITOKENS which follows the same result patterns from Figure 3. Note that FLEXITOKENS adapts well to unseen scripts because we use a script-agnostic boundary predictor as opposed to Ahia et al. (2024) which introduced the idea of equitable tokenization via script-specific boundary predictor for every language script included during pretraining. Also, compound or rare words (especially medical terms or foreign-origin words like "hypertrophic") are split into meaningful subwords, enabling the model to learn more meaningful representations.

**Impact of scaling each module:** We experimented with scaling FLEXITOKENS by adding more layers to the tokenization, language modeling, and upsampling module. Overall, we observe (see Figure 7 in Appendix D) that increasing our model's parameters by adding more layers to each module improves performance. We also note that the compression rate increases as we add more layers to the model. We speculate that this gain is because more layers allow the model to create richer representations prior to tokenization. We note that for the choice of which module gains the most from layer addition, increasing the LM module with 2 layers (2,14,2) outperforms adding more layers to other parts of the model (3,12,3). These results provide insightful directions for future research on scaling FLEXITOKENS.

**Relationship between compression and model performance:** We explore various configurations of $\alpha$ and how it impacts performance and show average results across all tasks in Table 4 (see Appendix D for a breakdown of performance on each language). As we scale the compression rate from 3× to 5× and 10×, we observe a slight decline in performance, indicating that too much forced compression may result in loss of information, hurting the model. We also speculate that this issue might be attributed to the small model size used in our main experiments. Recent work has argued that larger models can handle larger vocabularies better (Tao et al., 2024). Its analogue in our case is training a larger model with more layers in the tokenization module, which we show improves performance in FLEXITOKENS (3,12,3). Further scaling of both the tokenization and LM module are likely to further improve performance.

Table 4: Ablation for $\alpha$: Average Accuracy and Compression Results Across Multiple Languages

| Model | SIB-200 | WikiANN | Multi. Senti. | XNLI | ILI | Med. Abs. | Avg |
|---|---|---|---|---|---|---|---|
| | | | Accuracy | | | | |
| FLEXITOKENS 10x | 53.76 | 64.35 | **72.99** | 65.23 | 89.07 | 62.95 | 68.06 |
| FLEXITOKENS 5x | 71.16 | 64.92 | 72.54 | 65.48 | 89.28 | **63.47** | 71.14 |
| FLEXITOKENS 3x | **72.55** | **66.02** | 72.74 | **66.25** | **90.33** | 62.74 | **71.77** |
| | | | Compression Rate $\pm$ Std | | | | |
| FLEXITOKENS 10x | 28.89 ± 11.06 | 28.01 ± 14.14 | 27.41 ± 12.12 | 29.06 ± 8.55 | 38.80 ± 38.80 | 13.22 ± 2.15 | 27.56 ± 14.47 |
| FLEXITOKENS 5x | 10.72 ± 1.54 | 11.17 ± 3.69 | 11.25 ± 2.86 | 12.15 ± 1.76 | 14.82 ± 14.82 | 5.63 ± 0.33 | 10.96 ± 4.17 |
| FLEXITOKENS 3x | 6.19 ± 0.53 | 6.26 ± 1.33 | 6.17 ± 1.03 | 6.83 ± 0.60 | 8.35 ± 8.35 | 3.21 ± 0.15 | 6.17 ± 2.00 |

## 5 RELATED WORK

**Tokenizer-free language modeling**   Several works have explored the possibilities of training language models without relying on subword tokenization, instead representing text directly as a sequence of bytes (Xue et al., 2022; Al-Rfou et al., 2018; Wang et al., 2024; Limisiewicz et al., 2024) or pixels (Lotz et al., 2023; Rust et al., 2023; Salesky et al., 2023). To address the efficiency challenges of processing raw characters or byte sequences on tokenizer free LMs, alternative architectures have proposed to either segment byte sequences into fixed-length (Nawrot et al., 2022; Clark et al., 2022; Godey et al., 2022; Tay et al., 2022; YU et al., 2023) or dynamic segments Nawrot et al. (2023); Ahia et al. (2024); Pagnoni et al. (2024); Hwang et al. (2025). However, these models are pretrained with a fixed target compression rate, which limits their ability to adapt to shifts in data distribution. Moreso, the H-Net hierarchical transformer architecture (Hwang et al., 2025) has been found by prior works to be subpar to BPE and unstable during training (Main, 2025).

**Adapting tokenizers to new distributions**   There has been little research on adapting tokenizer-free LMs to new data distributions. Mofijul Islam et al. (2022) propose a character-based tokenizer by distilling segmentation information from heuristic-based subword tokenization. In contrast, several studies have explored adaptation strategies for subword tokenizers, both at inference time and during fine-tuning. For instance, prior work has shown that improved segmentation of large numbers can enhance performance on arithmetic tasks without retraining (Singh & Strouse, 2024; Sathe et al., 2025). In multilingual and domain-specific settings, various approaches have been proposed to adapt subword tokenizers during fine-tuning. These involve refining the tokenizer vocabulary with new tokens from the target distribution and initializing the corresponding embeddings to better capture linguistic and domain-specific characteristics (Park et al., 2021; Alabi et al., 2022; Minixhofer et al., 2022; Sachidananda et al., 2021; Liu et al., 2023). However, our experiments indicate that subword tokenizers often underperform in low-resource and non-Latin script languages due to over-segmentation.

## 6 CONCLUSION

We introduced FLEXITOKENS, a flexible, gradient-based tokenization approach that enables language models to adapt their segmentation patterns during finetuning. Unlike prior methods that enforce static or fixed compression rates, our method promotes dynamic tokenization aligned with the structure of the target distribution. Through multilingual and domain-diverse evaluations, FLEXITOKENS consistently reduces token over-fragmentation, improves downstream task performance, and achieves higher compression without sacrificing accuracy. Our results highlight the importance of adaptable tokenization strategies for building more efficient and generalizable language models.

REPRODUCIBILITY STATEMENT

To ensure the reproducibility of our work, we provided a detailed explanation of FLEXITOKENS in §2.2, including all our parameters and hyperparameters in multiple sections of this work. In Table 1, we report the exact values of $\alpha$ and $\sigma$ we used for computing the tokenization upper and lower bounds used in our experiments. We also described how we precomputed these values in §2.2. For easy comprehension of how our method works, we have expressed all our equations in familiar language modeling terms, which are widely known in the field. All the datasets we used for pretraining, fine-tuning, and evaluation are hosted on public platforms such as HuggingFace and GitHub, while we have also added appropriate citations for each dataset (§3.1). All hyperparameters of our model architecture and training runs are reported in §3 and §B. We include our code in this submission, and upon acceptance, we will release our code and training recipes to support reproducibility and foster adoption of FLEXITOKENS in future research.

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

# APPENDIX

## A    LIMITATIONS

Our limited computational budget prevents us from training larger models with more language on larger datasets. We anticipate the results will improve with scaling, potentially providing even higher compression. We leave this exploration to future work. While we aimed for diversity of languages and scripts in our experiments, we acknowledge that we do not cover a vast majority of linguistic diversity. But our methods are general, and we believe our results should translate to more languages. We also acknowledge a tradeoff between the performance and compression rate of the languages, with higher compression leading to a slight decline in performance, with some languages being more sensitive than others. FLEXITOKENS shares limitations of other segmentation methods in that it may not be suitable for languages where morphemes are discontinuous and vowels are interspersed between consonant roots for inflection or sometimes omitted, such as Semitic languages or other languages with Templatic morphologies.

## B    MODEL ARCHITECTURE AND HYPERPARAMETERS

In Figure 5, we present the hierarchical transformer design used in FLEXITOKENS. We also show the flow of an input sequence through all the modules in the model.

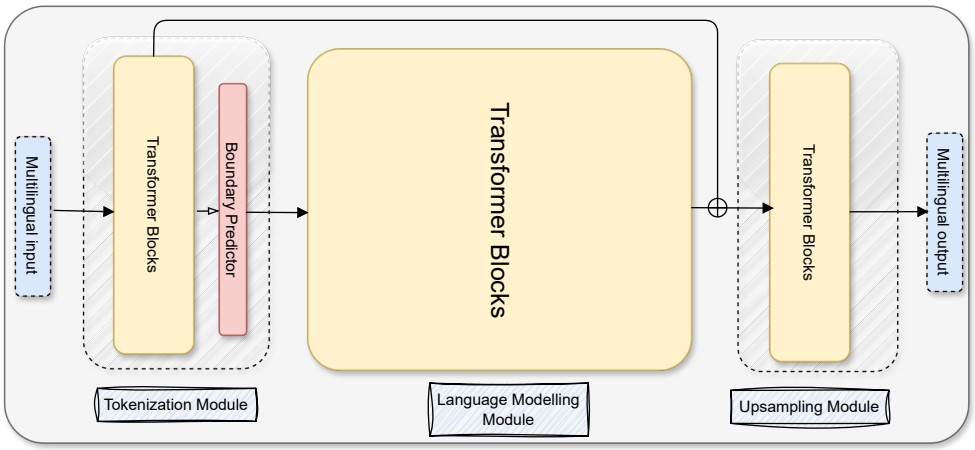

Figure 5: Hierarchical transformer design used in FLEXITOKENS. The input sequence is passed through the tokenization module for downsampling. The output of the downsampling phase is further processed by the language modeling module and finally upsampled back into the full sequence. Note that all stages use the same transformer block design.

**Finetuning Configurations and Hyperparameters**    During finetuning, we increase the sequence length to 2048 bytes to better capture longer sequences in the finetuning dataset.[9] For the NER task, we first concatenate token sequences using whitespaces before tokenization and label whitespaces as non-entity. We set gradient clipping to 1.0 and apply a warmup ratio of 10%. All tasks are finetuned for 5 epochs and an LR of 5e-5, except for OPUS-100 for which we finetuned for 2 epochs LR of 5e-4. We use task-specific batch sizes based on data availability. We perform monolingual finetuning on each language. Please refer to Table 5 for full finetuning parameters.

In Figure 6, we also show a distribution of the training dataset size we used for each language in our experiment's training corpus. In addition to English, we keep the number of samples for all other languages the same to avoid any bias that could be caused by data imbalance in our models.

---

[9]We use a shorter sequence length during pretraining due to computational constraints.

Table 5: Batch Sizes per Dataset and Language

| Dataset | en | es | ru | uk | hi | te | ur |
|---|---|---|---|---|---|---|---|
| XNLI | 64 | 64 | 64 | 64 | 64 | 64 | 64 |
| SIB-200 | 8 | 8 | 8 | 8 | 8 | 8 | - |
| WikiANN | 16 | 16 | 16 | 16 | 16 | 16 | - |
| Multi. Sentiment | 128 | 32 | 32 | - | 8 | - | - |
| Machine Transl. | - | 64 | 64 | - | 64 | - | - |
| Code-Switch | - | 32 | - | - | 32 | - | - |
| ILI | - | - | - | - | 32 | - | - |
| Medical Abstract | 16 | - | - | - | - | - | - |
| Irony detection | 32 | - | - | - | - | - | - |

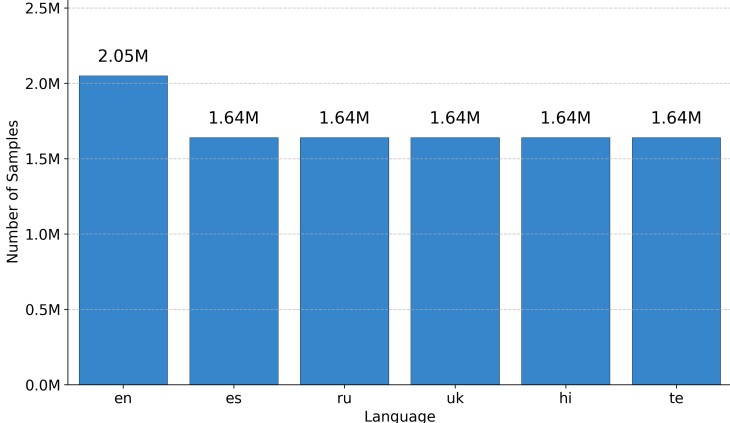

Figure 6: Number of training documents sampled by language

## C  RESULTS AND ANALYSES

In this section, we present the full results discussed in §4 across all our selected downstream tasks as seen in Table 7, 8, 9, and 10. We also present the full results for our multilingual sentiment analysis (Table 6) and SIB-200 (Table 8) evaluations . All Results in this section contain values for performance metrics like accuracy and F1 score, compression rates, and standard deviation of the compression rates.

Table 6: Multilingual Sentiment Accuracy and Compression Results for 3x Configurations

| Model | es | ru | hi | Avg |
|---|---|---|---|---|
| **Accuracy** | | | | |
| BPE | − | − | − | − |
| Binomial 3x | **77.89** | 87.20 | **53.63** | **72.91** |
| FLEXITOKENS $\lambda1$ | 77.75 | **87.33** | 53.42 | 72.83 |
| FLEXITOKENS $\lambda2$ | 77.77 | **87.33** | 53.12 | 72.74 |
| FLEXITOKENS $\lambda3$ | 77.63 | 87.13 | 53.01 | 72.59 |
| **Compression Rate ± Std** | | | | |
| Binomial | 3.61 ± 0.48 | 5.97 ± 0.98 | 7.98 ± 1.90 | 5.85 ± 1.27 |
| FLEXITOKENS $\lambda1$ | 3.78 ± 0.27 | 6.22 ± 0.53 | 8.26 ± 1.82 | 6.09 ± 1.11 |
| FLEXITOKENS $\lambda2$ | 3.90 ± 0.28 | 6.44 ± 0.61 | 8.16 ± 1.65 | 6.17 ± 1.03 |
| FLEXITOKENS $\lambda3$ | 4.04 ± 0.37 | 6.67 ± 0.75 | 8.83 ± 1.84 | 6.51 ± 1.17 |

Table 7: WikiANN NER F1 Score and Compression Results for 3x Configurations

| Model | en | es | ru | uk | hi | te | Avg |
|---|---|---|---|---|---|---|---|
| **F1 Score** | | | | | | | |
| BPE | 52.30 | 67.7 0 | 64.94 | 74.99 | 60.23 | 48.18 | 61.39 |
| Binomial | 63.80 | 75.06 | 67.59 | **78.06** | 61.21 | 48.31 | 65.67 |
| FLEXITOKENS $\lambda 1$ | 63.07 | 76.12 | **68.30** | 77.94 | **62.26** | **51.74** | **66.57** |
| FLEXITOKENS $\lambda 2$ | **63.96** | **76.23** | 67.55 | 77.99 | 62.24 | 48.13 | 66.02 |
| FLEXITOKENS $\lambda 3$ | 63.73 | 75.45 | 68.25 | 78.01 | 61.97 | 50.88 | 66.38 |
| **Compression Rate $\pm$ Std** | | | | | | | |
| Binomial 3x | $3.05 \pm 0.47$ | $3.88 \pm 0.76$ | $6.37 \pm 1.67$ | $5.75 \pm 1.11$ | $8.74 \pm 3.27$ | $8.56 \pm 2.29$ | $6.06 \pm 1.86$ |
| FLEXITOKENS $\lambda 1$ | $3.18 \pm 0.43$ | $3.84 \pm 0.54$ | $6.31 \pm 1.15$ | $5.92 \pm 0.90$ | $8.42 \pm 1.68$ | $8.64 \pm 1.55$ | $6.05 \pm 1.14$ |
| FLEXITOKENS $\lambda 2$ | $3.27 \pm 0.44$ | $3.93 \pm 0.58$ | $6.58 \pm 1.38$ | $6.12 \pm 1.00$ | $8.52 \pm 1.49$ | $9.15 \pm 2.21$ | $5.66 \pm 1.33$ |
| FLEXITOKENS $\lambda 3$ | $3.42 \pm 0.53$ | $4.18 \pm 0.66$ | $6.64 \pm 1.29$ | $6.30 \pm 1.07$ | $8.76 \pm 1.77$ | $8.99 \pm 2.07$ | $6.38 \pm 1.35$ |

Table 8: SIB-200 Accuracy and Compression Results for with 3x Configurations

| Model | en | es | ru | uk | hi | te | Avg |
|---|---|---|---|---|---|---|---|
| **Accuracy** | | | | | | | |
| BPE | **80.88** | **81.37** | **81.37** | **76.96** | 60.78 | **72.55** | **75.65** |
| Binomial | 79.41 | 74.02 | 71.08 | 68.63 | 64.71 | 69.61 | 71.24 |
| FLEXITOKENS $\lambda 1$ | 78.92 | 72.55 | 75.49 | 69.61 | 61.27 | 66.18 | 70.67 |
| FLEXITOKENS $\lambda 2$ | 77.94 | 75.98 | 74.51 | 71.57 | 69.12 | 66.18 | 72.55 |
| FLEXITOKENS $\lambda 3$ | **80.88** | 77.45 | 73.04 | 72.55 | **71.08** | 71.08 | 74.35 |
| **Compression Rate $\pm$ Std** | | | | | | | |
| Binomial | $3.04 \pm 0.27$ | $3.70 \pm 0.34$ | $5.97 \pm 0.64$ | $6.26 \pm 0.70$ | $6.59 \pm 0.48$ | $10.16 \pm 1.34$ | $5.95 \pm 0.72$ |
| FLEXITOKENS $\lambda 1$ | $3.13 \pm 0.25$ | $3.81 \pm 0.29$ | $6.35 \pm 0.64$ | $6.40 \pm 0.64$ | $8.46 \pm 0.82$ | $8.44 \pm 0.61$ | $6.10 \pm 0.58$ |
| FLEXITOKENS $\lambda 2$ | $3.32 \pm 0.27$ | $3.92 \pm 0.31$ | $6.49 \pm 0.56$ | $6.06 \pm 0.54$ | $8.35 \pm 0.54$ | $9.00 \pm 0.79$ | $6.19 \pm 0.53$ |
| FLEXITOKENS $\lambda 3$ | $3.34 \pm 0.35$ | $4.19 \pm 0.38$ | $6.55 \pm 0.75$ | $6.36 \pm 0.81$ | $8.36 \pm 0.59$ | $9.65 \pm 1.28$ | $6.41 \pm 0.76$ |

Table 9: XNLI Accuracy and Compression Results for 3x Configurations

| Model | en | es | ru | hi | te | ur (OOD) | Avg |
|---|---|---|---|---|---|---|---|
| **Accuracy** | | | | | | | |
| BPE | 73.09 | 69.9 | 65.95 | 61.48 | 68 | 54.11 | 65.42 |
| Binomial | 72.87 | 70.28 | 65.93 | 62.26 | 66.11 | 54.79 | 65.37 |
| FLEXITOKENS $\lambda 1$ | **73.51** | 70.22 | 66.47 | **62.42** | 67.11 | 56.99 | 66.12 |
| FLEXITOKENS $\lambda 2$ | 73.21 | **70.84** | **66.97** | 62.16 | 66.71 | **57.58** | **66.25** |
| FLEXITOKENS $\lambda 3$ | 73.35 | 70.22 | 66.75 | 62.36 | **67.82** | 57.33 | 66.31 |
| **Compression Rate $\pm$ Std** | | | | | | | |
| Binomial | $3.13 \pm 0.30$ | $3.79 \pm 0.48$ | $6.10 \pm 0.74$ | $9.85 \pm 1.28$ | $8.37 \pm 1.21$ | $8.58 \pm 0.82$ | $6.64 \pm 0.88$ |
| FLEXITOKENS $\lambda 1$ | $3.17 \pm 0.19$ | $3.89 \pm 0.26$ | $6.47 \pm 0.53$ | $7.99 \pm 0.75$ | $8.39 \pm 0.58$ | $8.52 \pm 0.71$ | $6.40 \pm 0.55$ |
| FLEXITOKENS $\lambda 2$ | $3.36 \pm 0.26$ | $4.10 \pm 0.30$ | $6.98 \pm 0.60$ | $9.18 \pm 0.85$ | $8.62 \pm 0.65$ | $8.73 \pm 0.73$ | $6.83 \pm 0.60$ |
| FLEXITOKENS $\lambda 3$ | $3.56 \pm 0.31$ | $4.32 \pm 0.34$ | $7.45 \pm 0.72$ | $10.06 \pm 1.17$ | $8.95 \pm 0.74$ | $9.07 \pm 0.80$ | $7.24 \pm 0.74$ |

# D    FULL ABLATION RESULTS

We present the full ablation results as discussed in §4 in Table 4. All results in this section (12, 13, 14, 15, and 16) contain values for performance metrics like accuracy and F1 score, compression rates, and standard deviation of the compression rates.

Table 10: ILI, Medical Abstracts, and Irony (for $3\times$ Configuration)

| Model | ILI (hi) | Med. Abs. (en) | Irony (en) |
|---|---|---|---|
| **Accuracy** | | | |
| BPE | 89.06 | 57.68 | 67.86 |
| Binomial | 89.47 | 62.81 | 67.60 |
| FlexiTokens $\lambda1$ | 89.58 | 62.92 | 68.37 |
| FlexiTokens $\lambda2$ | **90.33** | 62.74 | 68.75 |
| FlexiTokens $\lambda3$ | 89.55 | **63.19** | **69.26** |
| **Compression Rate $\pm$ Std** | | | |
| Binomial 3x | $8.02 \pm 1.38$ | $3.01 \pm 0.13$ | $3.05 \pm 0.14$ |
| FlexiTokens $\lambda1$ | $8.04 \pm 0.89$ | $3.11 \pm 0.13$ | $3.09 \pm 0.08$ |
| FlexiTokens $\lambda2$ | $8.35 \pm 0.87$ | $3.21 \pm 0.15$ | $3.22 \pm 0.31$ |
| FlexiTokens $\lambda3$ | $\mathbf{8.77} \pm 1.21$ | $\mathbf{3.43} \pm 0.18$ | $\mathbf{3.36} \pm 0.13$ |

Table 11: Accuracy on MMLU, COPA, and INCLUDE tasks

| Model | copa (en) | copa (es) | mmlu (en) | mmlu (hi) |
|---|---|---|---|---|
| **MMLU and COPA** | | | | |
| BPE | 55.00 | 49.20 | 23.50 | 24.30 |
| Binomial | 55.00 | 50.40 | 23.00 | 24.30 |
| FlexiTokens $\lambda3$ | **56.00** | **54.40** | **23.30** | **24.80** |

| Model | hi | ru | es | te | uk |
|---|---|---|---|---|---|
| **INCLUDE** | | | | | |
| BPE | 25.05 | 26.99 | **27.45** | 24.09 | 33.64 |
| Binomial | 23.77 | **25.18** | 28.00 | 23.72 | 34.55 |
| FlexiTokens $\lambda3$ | 23.95 | 25.72 | 26.55 | 23.91 | **35.09** |

Table 12: SIB-200 $\alpha$ Ablation: Accuracy and Compression Results

| Model | en | es | ru | uk | hi | te | Avg |
|---|---|---|---|---|---|---|---|
| **Accuracy** | | | | | | | |
| FlexiTokens 10x | 57.35 | 59.80 | 55.88 | 50.98 | 47.06 | 51.47 | 53.76 |
| FlexiTokens 5x | **78.92** | **78.92** | 74.51 | **73.04** | 62.75 | 58.82 | 71.16 |
| FlexiTokens 3x | 77.94 | 75.98 | **74.51** | 71.57 | **69.12** | **66.18** | **72.55** |
| **Compression Rate $\pm$ Std** | | | | | | | |
| FlexiTokens 10x | $19.37 \pm 8.23$ | $16.23 \pm 4.45$ | $24.57 \pm 6.82$ | $28.69 \pm 8.88$ | $40.06 \pm 14.68$ | $44.43 \pm 17.47$ | $28.89 \pm 11.06$ |
| FlexiTokens 5x | $5.75 \pm 0.65$ | $6.78 \pm 0.71$ | $12.58 \pm 1.91$ | $10.62 \pm 1.70$ | $13.42 \pm 1.63$ | $15.17 \pm 2.04$ | $10.72 \pm 1.54$ |
| FlexiTokens 3x | $3.32 \pm 0.27$ | $3.92 \pm 0.31$ | $6.49 \pm 0.56$ | $6.06 \pm 0.54$ | $8.35 \pm 0.54$ | $9.00 \pm 0.79$ | $6.19 \pm 0.53$ |

Table 13: WikiANN $\alpha$ Ablation: F1 Score and Compression Results

| Model | en | es | ru | uk | hi | te | Avg |
|---|---|---|---|---|---|---|---|
| **F1 Score** | | | | | | | |
| FLEXITOKENS 10x | 61.81 | 75.48 | 66.90 | 76.90 | 59.88 | 45.15 | 64.35 |
| FLEXITOKENS 5x | 62.84 | 75.81 | 67.48 | 77.68 | 60.02 | 45.66 | 64.92 |
| FLEXITOKENS 3x | **63.96** | **76.23** | **67.55** | **77.99** | **62.24** | **48.13** | **66.02** |
| **Compression Rate $\pm$ Std** | | | | | | | |
| FLEXITOKENS 10x | $14.15 \pm 6.07$ | $16.87 \pm 6.39$ | $40.03 \pm 19.10$ | $27.91 \pm 11.95$ | $42.52 \pm 21.82$ | $26.55 \pm 11.73$ | $28.01 \pm 14.14$ |
| FLEXITOKENS 5x | $5.83 \pm 1.23$ | $7.26 \pm 2.01$ | $15.30 \pm 5.90$ | $11.93 \pm 3.59$ | $15.92 \pm 4.68$ | $10.80 \pm 2.57$ | $11.17 \pm 3.69$ |
| FLEXITOKENS 3x | $3.27 \pm 0.44$ | $3.93 \pm 0.58$ | $8.52 \pm 1.49$ | $6.58 \pm 1.38$ | $9.15 \pm 2.21$ | $6.12 \pm 1.00$ | $6.26 \pm 1.33$ |

Table 14: XNLI $\alpha$ Ablation: Accuracy and Compression Results

| Model | en | es | ru | hi | te | ur | Avg |
|---|---|---|---|---|---|---|---|
| **Accuracy** | | | | | | | |
| FLEXITOKENS 10x | 71.42 | 68.60 | 65.59 | 62.22 | 66.05 | 57.52 | 65.23 |
| FLEXITOKENS 5x | 72.97 | 70.38 | 65.47 | 61.88 | 65.49 | 56.71 | 65.48 |
| FLEXITOKENS 3x | **73.21** | **70.84** | **66.97** | 62.16 | **66.71** | **57.58** | **66.25** |
| **Compression Rate $\pm$ Std** | | | | | | | |
| FLEXITOKENS 10x | $13.41 \pm 2.88$ | $15.88 \pm 3.12$ | $25.20 \pm 6.07$ | $41.81 \pm 12.06$ | $37.23 \pm 8.77$ | $40.84 \pm 12.71$ | $29.06 \pm 8.55$ |
| FLEXITOKENS 5x | $6.06 \pm 0.72$ | $7.59 \pm 0.88$ | $13.02 \pm 2.08$ | $15.44 \pm 2.16$ | $15.10 \pm 1.60$ | $15.67 \pm 2.40$ | $12.15 \pm 1.76$ |
| FLEXITOKENS 3x | $3.36 \pm 0.26$ | $4.10 \pm 0.30$ | $6.98 \pm 0.60$ | $9.18 \pm 0.85$ | $8.62 \pm 0.65$ | $8.73 \pm 0.73$ | $6.83 \pm 0.60$ |

Table 15: Multilingual Sentiment $\alpha$ Ablation: Accuracy and Compression Results

| Model | es | ru | hi | avg |
|---|---|---|---|---|
| **Accuracy** | | | | |
| FLEXITOKENS 10x | 77.67 | 87.07 | **54.24** | **72.99** |
| FLEXITOKENS 5x | 77.74 | 87.17 | 52.71 | 72.54 |
| FLEXITOKENS 3x | **77.77** | **87.33** | 53.12 | 72.74 |
| **Compression Rate $\pm$ Std** | | | | |
| FLEXITOKENS 10x | $16.07 \pm 4.53$ | $26.55 \pm 8.33$ | $39.60 \pm 21.99$ | $27.41 \pm 12.12$ |
| FLEXITOKENS 5x | $6.83 \pm 0.77$ | $11.45 \pm 2.11$ | $15.47 \pm 5.21$ | $11.25 \pm 2.86$ |
| FLEXITOKENS 3x | $3.90 \pm 0.28$ | $6.44 \pm 0.61$ | $8.16 \pm 1.65$ | $6.17 \pm 1.03$ |

Table 16: ILI (hi) and Medical Abstract (en) $\lambda$ Ablation: Accuracy and Compression Results

| Model | ILI (hi) | Med. Abstract (en) |
|---|---|---|
| **Accuracy** | | |
| FLEXITOKENS 10x | 89.07 | 62.95 |
| FLEXITOKENS 5x | 89.28 | **63.47** |
| FLEXITOKENS 3x | **90.33** | 62.74 |
| **Compression Rate $\pm$ Std** | | |
| FLEXITOKENS 10x | $38.80 \pm 16.75$ | $13.22 \pm 2.15$ |
| FLEXITOKENS 5x | $14.82 \pm 3.00$ | $5.63 \pm 0.33$ |
| FLEXITOKENS 3x | $8.35 \pm 0.87$ | $3.21 \pm 0.15$ |

We also present the results of our ablation study on how scaling the model's tokenization module impacts performance in Figure 7

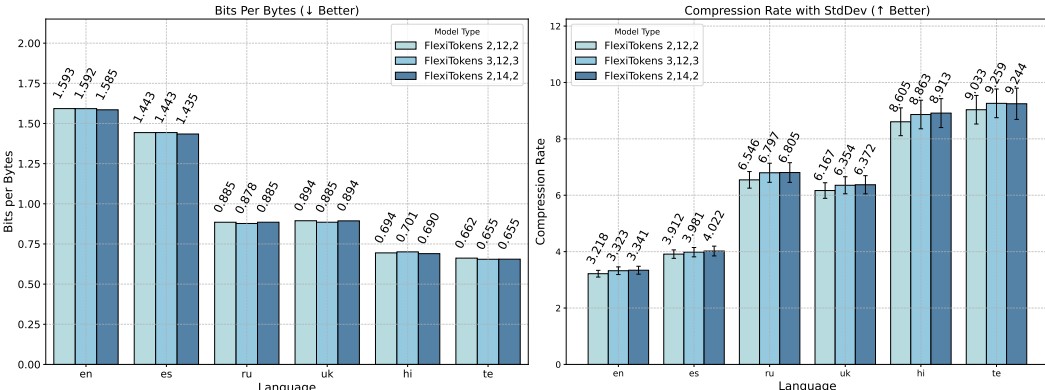

Figure 7: FineWeb Test results for ablating the number layers in FLEXITOKENS. Adding more layers results to lower BPB and higher compression rate across all model sizes. FLEXITOKENS (2,12,2) is equivalent to 2, 12 and 2 transformer layers in the tokenization, LM, and upsampling module, respectively.

Table 17: Tokenization outputs with different methods (Urdu, Telugu, English)

| Tokenizer | Sentence and Segmentation | #Tokens |
|---|---|---|
| **ur** | 39-year-old SpongeBob was diagnosed with hypertrophic cardiomy-opathy in Mumbai. | – |
| BPE | 39Ø³ØŞÙÛġ ØŞØ³Ù¾ÙĨØ¬ Ø¨ØŞØ¨ Ú©ÙĪ ÙÙØ¨Ø¦ÛĮ ÙÛĮÚº Ûġ ØŞÛĮÙ¾Ø±Ù¹Ø±ØŞÙġÚ© Ú©ØŞØ±Ú¦ÛÛĮÙÛĮÙÛÛ¾ÛĮØ ªÚ¾ÛĮ Ú©©ÛĮ Ø ªØ´ Ø®ÛĮØµ Ûġ ÙÛĮ¦ÛĮÛĶ | 107 |
| BINOMIAL 3× | 3ǀ9 ǀ ǀ ǀ ǀ ǀǀ ǀ ǀ ǀǀǀ ǀǀǀ ǀ ǀǀ ǀǀ | 21 |
| FLEXITOKENS 3× | 39 ǀ ǀ ǀ ǀ ǀ ǀǀ ǀ ǀǀǀǀǀ ǀ ǀ ǀ | 17 |
| **te** | He spent the whole night watching Netflix. He fell asleep early. | – |
| BPE | à°à°¤à°¿ǀà�±ġǀ à°°ǀà°¾à°¤ǀà�±à°°ǀà°Ħǀà°¾ǀà°¾ǀ à°¨ǀà�±Īà°Ł ǀà౱ǀâGǀǀĮà°«ǀà౱à°²ǀà°¿ǀà°ºkǀà౱à°ǀ ǀà౱à°ǀ à°ǀǀà°Ħ à°ǀǀà౱à°¤ǀà°Ħ ǀ à°Łà°¿ǀà°¿ǀà౰ªǀà°¾à°¾à°¿ǀà౱à°ġ. à°à°¤à°¿ǀà౱ġǀ à°¤ǀà౱à°ºà°µà°°à°Ł ǀà°¾à°¾ǀ à°¨ǀǀà°¿ǀà°¾à°°à౰ªǀà±à°±à౰ ¯à°¾à°¾ǀà౱à°ġ. | 37 |
| BINOMIAL 3× | ǀǀ ǀǀ ǀǀǀ ǀ ǀǀ. ǀ ǀǀ ǀǀǀ. | 22 |
| FLEXITOKENS 3× | ǀ ǀ ǀǀǀǀ ǀǀ :. ǀ ǀ ǀǀǀ:. | 17 |
| **en** | Influenza and pneumonia were identified as major causes of mortality in children. | – |
| BPE | `Influenza and pneumonia were identified as major` `causes of mortality in children.` | 20 |
| BINOMIAL 3× | `Influenza and pneumonia were identified as major` `causes of mortality in children.` | 25 |
| FLEXITOKENS 3× | `Influenza and pneumonia were identified as major` `causes of mortality in children.` | 20 |
| **en** | Oh no, another surprise bonus at work. Just what I didn't need grinning-faceface-with-tears-of-joyface-with-tears-of-joyman-raising-hand-medium-skin-tonewoman-cartwheeling-medium-dark-skin-tone. | – |
| BPE | Ohǀ noǀ,ǀ anotherǀ surpriseǀ bonusǀ atǀ workǀ.ǀ Justǀ whatǀ Iǀ didnǀâGĿtǀ needǀ ðĿǀGǀðĿĴĤĮðĿĴĤĮðĿĿĮǀÏǀðĿıǀ½âGǀǀâĿĮĤĮǀǀðĿıǀ¤ǀðĿıǀ¾âGǀ ǀǀâĿǀGǀÏǀıǀħǀǀ§ĦǀkĦǀǀĲĦǀǀijĦǀǀ¢ĦǀǀkĦǀǀĲĦǀǀĲĦǀǀ█ĦǀǀĿĦǀǀĿĦǀǀĿĦǀĿǀǀǀkĦǀǀKǀ. | 82 |
| BINOMIAL 3× | `Oh no, ǀanǀotǀherǀ ǀsurǀprǀise ǀbonǀus ǀatǀ` `ǀwǀork. ǀJustǀ ǀwǀhat ǀI didn'ǀtǀ ǀneed ǀ` `grinning-faceǀǀǀǀǀface-with-tears-of-joyface-with-tears-of-joyǀ` `man-raising-hand-medium-skin-toneǀǀǀǀwoman-cartwheeling-medium-dark-skin-toneǀ.` | 33 |
| FLEXITOKENS 3× | `Oh ǀno, ǀanotǀher ǀsurǀprǀise ǀbonus ǀatǀ ǀworǀk.ǀ` `ǀJǀust ǀwhatǀ ǀI ǀdidnǀtǀ ǀneed ǀ` `grinning-faceface-with-tears-of-joyface-with-tears-of-joyǀ` `man-raising-hand-medium-skin-toneǀ` `woman-cartwheeling-medium-dark-skin-toneǀ.` | 25 |