# OpenReview forum: "FLEXITOKENS: Flexible Tokenization for Evolving  Multilingual Language Models"
_ICLR.cc/2026/Conference — ICLR 2026 Conference Withdrawn Submission_

### Official Review · Reviewer_81UQ · 2025-10-27

**Soundness:** 3
**Presentation:** 3
**Contribution:** 2
**Rating:** 4
**Confidence:** 3

**Summary:**

The authors build upon prior work on LMs with gradient-based tokenization and address the issue of fixed compression rates by replacing the binomial loss with a new loss function. This modification enables the model to dynamically adjust its compression rate, thereby reducing over-fragmentation and improving downstream task performance.

**Strengths:**

1.The paper is well structured and highly readable.

2.The motivation is clearly stated, and the methodology is explained in detail.

**Weaknesses:**

1.The experimental section of the paper includes only two baselines: BPE and Binomial loss, which are insufficient to fully demonstrate the effectiveness of the proposed method.

2.Most of the experiments are conducted on a model with 119M parameters, making it unclear whether the proposed approach can maintain its effectiveness when scaled to larger language models.

**Questions:**

1.It would be helpful if the paper included a diagram or illustration of the proposed method in the main text, so that readers could more easily grasp the overall framework.

2.The proposed method requires a small n-way parallel corpus to compute per-language parameters. Could the authors clarify whether the choice of this parallel corpus has an impact on the resulting compression rates or model performance?

3.The experiments include only one gradient-based tokenization baseline. By the way,can this method be applied to different model architectures beyond the one used in this paper?

---

> ### Author Response · Authors · 2025-11-20
>
> We thank the reviewer for their feedback and questions. Please find our responses to your questions and concerns below:
> $~$
>
> > **The experimental section of the paper includes only two baselines...**
>
> We only include these baselines as they are the most comparable methods to our work on flexible tokenization. This is because our core contribution is a new training objective that allows tokenization to change as you change domains. Our training objective can therefore be applied to other similar hierarchical transformer models, while BPE is a standard reference for fixed tokenization. Our contribution is similar to that of SigLIP (Tschannen et al., 2025) over CLIP (Radford et al., 2021), where they majorly introduced a new training objective. Similarly, BYOL (Grill et al., 2020), SimSiam (Chen et al., 2021), and Barlow Twins (Zbontar et al., 2021) all differ from SimCLR (Chen et al., 2020) by just using different training objectives.
>
> $~$
>
> > **Most of the experiments are conducted on a model with 119M parameters…**
>
> We used a model with a 119M parameter size because we were constrained by our academic budget. Each pretraining run takes 60 hours on an H100 GPU, not including the extensive post-training required for adaptation across multiple languages, domains, and tasks. In total, we conducted over 250 training runs for this paper alone (in addition to many exploratory runs that were not reported), which restricted us to smaller model sizes. Previous works have also used models with smaller parameters sizes (44M to 100M) to demonstrate the effectiveness of their methods (Nawrot et al., 2023; Fleshman & Van Durme, 2023; Ahia et al., 2024).
>
> $~$
>
> > **It would be helpful if the paper included a diagram or illustration of the proposed method in the main text...**
>
> We already include this in our work. Please see Figure 5 in Appendix B for the structure of our model. Given more space in the final version, we will move the figure to the main paper.
>
> $~$
> > **The proposed method requires a small n-way parallel corpus …**
>
> Any n-way dataset with a reasonably uniform sentence-length distribution can be used in our method. This is because our formula for computing compression rates ensures that if the reference language’s rate shifts, the rates for all other languages shift proportionally. In other words, the relative compression pattern is preserved regardless of which parallel corpus is used. To show this, we report $\beta$ values computed on FLORES and MGSM. As the $\beta$ value for English increases, the values for the other languages follow the same trend, keeping tokenization fair across languages:
>
> | Language | FLORES | MGSM |
> |----------|--------|-------|
> | en       | 0.356  | 0.363 |
> | es       | 0.294  | 0.332 |
> | ru       | 0.178  | 0.199 |
> | te       | 0.132  | 0.134 |
>
> $~$
>
> > **The experiments include only one gradient-based tokenization baseline. By the way, can this method be applied to different model architectures beyond the one used in this paper?**
>
> Yes, our method applies to other model architectures, as our major contribution is a new training objective that enhances tokenization flexibility and not a new gradient-based architecture.
>
> $~$
>
> **References**
>
> 1. Nawrot, P., Chorowski, J., Łańcucki, A. and Ponti, E.M., 2022. Efficient transformers with dynamic token pooling. arXiv preprint arXiv:2211.09761.
>
> 2. Fleshman, W. and Van Durme, B., 2023. Toucan: Token-aware character level language modeling. arXiv preprint arXiv:2311.08620.
>
> 3. Ahia, O., Kumar, S., Gonen, H., Hofmann, V., Limisiewicz, T., Tsvetkov, Y. and Smith, N.A., 2024. Magnet: Improving the multilingual fairness of language models with adaptive gradient-based tokenization. *Advances in Neural Information Processing Systems*, 37, pp.47790–47814.
>
> 4. Tschannen, Michael, et al. "Siglip 2: Multilingual vision-language encoders with improved semantic understanding, localization, and dense features." arXiv preprint arXiv:2502.14786 (2025).
>
> 5. Zbontar, Jure, et al. "Barlow twins: Self-supervised learning via redundancy reduction." *International Conference on Machine Learning*. PMLR, 2021.
>
> 6. Radford, Alec, et al. "Learning transferable visual models from natural language supervision." *International Conference on Machine Learning*. PMLR, 2021.
>
> 7. Chen, Xinlei, and Kaiming He. "Exploring simple siamese representation learning." *Proceedings of the IEEE/CVF Conference on Computer Vision and Pattern Recognition*. 2021.
>
> 8. Grill, Jean-Bastien, et al. "Bootstrap your own latent—a new approach to self-supervised learning." *Advances in Neural Information Processing Systems* 33 (2020): 21271–21284.
>
> 9. Chen, Ting, et al. "A simple framework for contrastive learning of visual representations." *International Conference on Machine Learning*. PMLR, 2020.

---

> > ### Comment · Reviewer_81UQ · 2025-11-27
> >
> > Thanks to the authors for their efforts and responses. However, some concerns remain unresolved. For example, regarding the response to the last question, providing the corresponding experimental results would better support the authors’ claims.

---

### Official Review · Reviewer_vKnb · 2025-10-30

**Soundness:** 1
**Presentation:** 2
**Contribution:** 2
**Rating:** 2
**Confidence:** 4

**Summary:**

This paper addresses the problem of overfragmentation in subword tokenization. To remedy this, the paper proposes using byte-level LMs with a boundary predictor. (i.e., internally, a model processes a sequence by segment instead of byte.) The proposed method also introduces an additional loss term to make the single boundary predictor effective across different languages. The experiments are conducted on a small-scale setup (119M); the proposed method is compared with the standard BPE-based model and a variant without the additional loss term. For downstream evaluation, each model is fine-tuned on target-domain data. The results show that the proposed method (especially $\lambda$3) outperforms baselines in terms of encoding efficiency and downstream performance.

**Strengths:**

1. The paper proposes a novel method for encoding and processing multilingual text more effectively and efficiently. The methodology is well-motivated and relatively easy to understand.
2. The results of a small-scale experiment show promising results over the standard BPE baseline and the baseline: Binomial, which does not employ the proposed flexible loss function, in terms of encoding efficiency and fine-tuned downstream performance (when used with $\lambda3$).

**Weaknesses:**

Overall, the paper proposes a novel method to encode/process multilingual text more effectively and efficiently. Nonetheless, its experimental design and the discussion/analysis require substantial modification.

1. Some expressions in the paper are quite awkward and require a fix. For instance, “Language models are challenging to adapt… (L11)” is grammatically unnatural and should be modified as something like: “Adapting language models to new data distributions by simple finetuning is challenging”. The same applies to “overfragmentation of out-of-distribution domains (L14)”.

2. The major limitation of this paper is a lack of state-of-the-art baselines. This paper deliberately omits a comparison with the closest related work, HNet (Hwang et al., 2025; https://arxiv.org/abs/2507.07955), citing that it is unstable during training (Main, 2025; https://main-horse.github.io/hnet/scaling-laws-byte/). This is a rather weak argument, and citing a blog post to support the claim is not entirely reliable. Also, other studies address a similar challenge (e.g., the Byte Latent Transformer; ACL 2025; https://aclanthology.org/2025.acl-long.453.pdf). The lack of such state-of-the-art baselines prevents us from evaluating the actual impact/effectiveness of the proposed method in comparison to them.

3. Another major limitation of this paper is the use of a small-scale model (119M). Due to this limitation, most of the presented results are from fine-tuning. This is an uncommon practice in the previous literature, where they use zero-shot and few-shot evaluation. Also, this small scale appears to favor the proposed method over the BPE baseline, as the BPE baseline has only five layers compared to 16 for the proposed method. On a larger scale, the observed trend might substantially change.

4. Some results are missing without justification. For instance, the results for the proposed method ($\lambda$1) are missing for CS en-es/CS en-hi in Table 3. Table 2 does not have the results for $\lambda$1 and $\lambda$2. In particular, L323-L358 requires corresponding results for $\lambda$1 and $\lambda$2 to be more convincing. The current statement lacks clear evidence and is not convincing, given that a higher $\lambda$ does not always achieve the best results in Table 3. Furthermore, why does the MT evaluation in Table 2 only cover three language pairs?

5. The mentioned qualitative analysis in L366-369 is missing.

**Questions:**

1. On Weakness 1, please make sure to make necessary changes regarding the overall writing of the submission.
2. In Section 2.1, how does “the language modelling module” pool the byte-level hidden states? Is it just mean pooling?
3. On Weakness 2, I would suggest including HNet and BLT baselines.
4. On Weakness 3, I would suggest including a larger-scale experiment (e.g., at least 1B).
5. On Weaknesses 4 and 5, please make sure to include the missing results and fix the discussion accordingly.

---

> ### Author Response · Authors · 2025-11-20
>
> We thank the reviewer for their feedback and questions. Please find our responses to your questions and concerns below:
>
> $~$
>
> > **Some expressions in the paper are quite awkward and require a fix…**
>
> We will reflect these suggested changes in our revised version.
>
> $~$
>
> > - **The major limitation of this paper is a lack of state-of-the-art baselines …**
> > - **On Weakness 2, I would suggest including HNet and BLT baselines.**
>
> We would like to clarify that our contribution is not a new architecture but a new training objective for boundary prediction. This objective can be applied to any end-to-end dynamic-tokenization model, including HNet. Both HNet and DTP (Nawrot et al., 2022) operate in the same way: they use an Hourglass hierarchical transformer that predicts token boundaries. We chose the Hourglass architecture of DPT because its training procedure is well-established in prior work (Nawrot et al., 2023; Fleshman & Van Durme, 2023; Ahia et al., 2024), and its training resources are publicly available. In contrast, HNet is very recent, not yet published, and does not provide training code or recipes. The only difference between HNet and our implementation is the boundary-prediction loss. HNet uses a loss that enforces a fixed compression ratio similar to DTP and MAGNET. FlexiTokens introduces a relaxed objective that allows the segmentation pattern to change when adapting to a new dataset. Applying our method to HNet would simply require swapping their boundary-loss term for ours; no architectural changes are needed.
>
> Regarding BLT (Byte Latent Transformer), it is **not** an end-to-end dynamic tokenization model. It relies on a separate learned tokenizer and latent sequence, so it cannot be directly compared to methods like ours that jointly train the tokenizer and language model.
>
> $~$
>
> > - **Another major limitation of this paper is the use of a small-scale model (119M)…**
> > - **On Weakness 3, I would suggest including a larger-scale experiment (e.g., at least 1B).**
>
> We used a model with a 119M parameter size because we were constrained by our academic budget. Each pretraining run takes 60 hours on an H100 GPU, not including the extensive post-training required for adaptation across multiple languages, domains, and tasks. In total, we conducted over 250 training runs for this paper alone (in addition to many exploratory runs that were not reported), which restricted us to smaller model sizes.
>
> We would also like to clarify that we did not reduce the parameter size of the BPE model; we only transferred parameters to the BPE model’s embedding table and ensured that the total parameters match those of FlexiTokens. This is how previous works have compared BPE models to byte-level models (Slagle et al., 2024).
>
> $~$
>
> > - **Some results are missing without justification.**
> > - **On Weaknesses 4 and 5, please make sure to include the missing results and fix the discussion accordingly.**
>
> We only exclude 8 out of over 120 results for $\lambda1$ and $\lambda2$ combined from some of our results, as we’ve clearly seen that $\lambda3$ consistently outperforms all our baselines. Please see Tables 6, 7, 8, 9, and 10 for extensive results of $\lambda1$ and $\lambda2$. Every other result not reported was because we did not have access to the evaluation datasets in those languages at the time of running our experiments.
>
> $~$
>
> > **The mentioned qualitative analysis in L366–369 is missing.**
>
> Please see Table 17 for the reference to our qualitative analysis.
>
> $~$
>
> > **In Section 2.1, how does “the language modelling module” pool the byte-level hidden states? Is it just mean pooling?**
>
> Yes, it is just mean pooling.
>
> We will reflect all these changes to the paper structure of our paper, which you have pointed out.
>
> $~$
>
> **References**
>
> 1. Nawrot, P., Chorowski, J., Łańcucki, A. and Ponti, E.M., 2022. *Efficient transformers with dynamic token pooling.* arXiv preprint arXiv:2211.09761.
> 2. Fleshman, W. and Van Durme, B., 2023. *Toucan: Token-aware character level language modeling.* arXiv preprint arXiv:2311.08620.
> 3. Ahia, O., Kumar, S., Gonen, H., Hofmann, V., Limisiewicz, T., Tsvetkov, Y. and Smith, N.A., 2024. *MAGNET: Improving the multilingual fairness of language models with adaptive gradient-based tokenization.* NeurIPS 37, pp. 47790–47814.
>
> 4. Slagle, Kevin. "Spacebyte: Towards deleting tokenization from large language modeling." Advances in Neural Information Processing Systems 37 (2024): 124925-124950.

---

> > ### Comment · Reviewer_vKnb · 2025-11-25
> >
> > Thanks for the response.
> >
> > On Weaknesses 2 and 3, I am not convinced by the response. The HNet paper is out July 10th, well before the ICLR submission deadline. The response does not address my concern about citing the blog post as a counterargument regarding not making the comparison against HNet. Moreover, if the only difference between HNet and this work lies in the loss term, why not compare them? This will effectively show the efficacy of the proposed boundary prediction objective. Moreover, it is important to compare the performance of the proposed method against methods in a similar paradigm as a reference (even if it is a bit challenging to conduct an apples-to-apples comparison). This will allow us to keep track of the progress in the domain. While I understand the resource constraints, the value of the work should be evaluated on the basis of the content, not the effort spent.
> >
> > Regarding Weakness 4, I do not think the response addresses it. The review says “the current statement lacks clear evidence and is not convincing given that a higher $\lambda$ does not always achieve the best results in Table 3”. The response does not provide concrete justifications for this.
> >
> > Overall, the response does not address my concerns. Therefore, I will maintain my score.

---

### Official Review · Reviewer_WiM7 · 2025-10-31

**Soundness:** 3
**Presentation:** 3
**Contribution:** 2
**Rating:** 4
**Confidence:** 4

**Summary:**

Tokenization with a fixed vocabulary (e.g., BPE) often suffers from rigidity, leading to over-fragmentation when applied to unseen languages or scripts. Recent work introduces a tokenization module within the language model that segments bytes into patches, learning the desired compression rate through gradients from an auxiliary loss. This is done by incorporating a tokenization module consisting of a Boundary Predictor and an upsampling module, replacing the standard vocabulary-bounded embedding and unembedding layers. The entire model, including these modules, is pretrained jointly.

However, the authors argue that using a predetermined compression rate introduces another form of rigidity.
To address this, the paper proposes a new auxiliary loss function, "FlexiToken" loss, a hinge-like objective that encourages the compression rate to remain within a range (b - a) rather than regressing to a single target value a (referred to as "Binomial" in prior work). In practice, this relaxes the existing training loss by introducing a lower bound and by not penalizing the tokenizer when the compression rate exceeds this threshold.
This is to design a better, flexible adaptation to new domains, languages, and scripts.
Additionally, the authors simplify the model by using a single Boundary Predictor shared across languages, instead of maintaining separate predictors for each language.

Main findings :
1. After pretraining, the proposed method achieves higher compression rates and greater variance in compression compared to the binomial baseline, while maintaining similar perplexity.
2. After fine-tuning on downstream tasks, the model with FlexiToken shows consistent performance improvements over the binomial baseline.

**Strengths:**

1. Comprehensive overview of current trends:
The paper provides a well-structured summary of recent advances in byte-level tokenization, clearly outlining existing approaches and highlighting their key limitations such as fixed compression rates and lack of adaptability across languages and scripts.
2. Clear motivation for the proposed method:
The motivation behind introducing a new loss function is well articulated, effectively connecting the rigidity of existing compression-based tokenization methods to the need for a more flexible framework.
3. Extensive experimental validation:
The paper has thorough experiments across a wide range of tasks and diverse languages. After fine-tuning on downstream tasks, the model with FlexiToken shows consistent performance improvements over the binomial baseline.
4. Well-formulated and technically sound proposal:
The introduction of the hinge-like FlexiToken loss is conceptually clear and mathematically grounded.

**Weaknesses:**

1. While the paper's central claim emphasizes improved flexibility and adaptability over the binomial baseline, the empirical evidence supporting this is not entirely convincing. In particular, the variance results discussed in L301 and illustrated in Figure 2 do not clearly demonstrate the reported increase in variance. Moreover, the observed improvements in compression rate may partially stem from the broader allowable range introduced by the proposed loss, rather than from intrinsic flexibility of the model itself.

2. Although the proposed method shows higher performance than the binomial baseline, it remains unclear whether these improvements arise specifically from the relaxation of the compression rate range or from other factors, like single vs separate boundary predictors. Further ablation study would strengthen this claim, for example, by fixing a single language and gradual increase of the lambda (e.g. 0, 0.1, 0.2, 0.4, 0.8, 1.6, 3.2) while keeping the same alpha as binomial experiment.

3. The comparison between the binomial and FlexiToken models' compression rates after fine-tuning (Figure 4) is not entirely convincing for demonstrating FlexiToken's advantage over binomial. While the first-row results for en, es, and ru show that the binomial model remains relatively low and stable compared to FlexiToken, the second-row results for hi, te, and uk suggest that the binomial model also adapts to the tasks to a similar extent. This seems like the binomial model, despite its "rigid" compression rate, is still capable of adapting to different tasks and languages.

4. (writings)
- Line 291: Bits per Byte (BPB) may need a brief explanatory phrase clarifying its relation to the original concept, as BPB is not mentioned in the original reference.
- Figure 2 (left, y-axis and titles) uses the Bits per Character (BPC), not BPB.
- Duplicate citations: Lines 702 and 706 contain repeated references that should be consolidated.
- Table 17: The BPE outputs are not readable.
- Figure 4: The x-axis ticks for es and ru are misaligned and should be corrected for consistency.
- It would be helpful to include a pointer to a specific example (if any in Appendix) that substantiates the qualitative observations (Lines 367–372) discussed in this section.

**Questions:**

1. Figure 1: It may make more sense to compare the proposed method with the binomial baseline rather than BPE. How does the binomial model fragment this text?
2. L156: The calculation of σ is not clearly explained. What exactly does it represent, and what is the tokenization rate before the FlexiToken module is trained?
3. Confused by these two non-penalization descriptions in the intro and methods section:
- L053: "By not penalizing the tokenizer when the compression rate is higher than …" suggests that penalization would push the compression rate lower.
- L158: "This loss will become 0, reducing further incentive to compress by not penalizing the model." implies that penalization would push for compression.

---

> ### Author Response · Authors · 2025-11-21
>
> We thank the reviewer for their feedback and questions. Please find our responses to your questions and concerns below:
>
> $~$
>
> >**While the paper's central claim emphasizes improved flexibility and adaptability over the binomial baseline, the empirical evidence supporting this is not entirely convincing…**
>
> The increased variance referred to in L301 is with regard to the general observations we found in our work beyond Figure 2, which uses a 3x compression rate.  In other experiments we performed with over 3x compression rate, the increase in variance becomes even more evident. Please see the table below for the results of these experiments with a 5x compression rate. Here, we can clearly see that Flexitokens continuously higher variance. This result was mistakenly omitted, and we will update the revised version of our paper with it.
>
> $~$
>
> **XNLI-en Results**
>
> | Model    | Acc.  | Comp. rate (Before/After) | Var. (Before/After) |
> |----------|-------|---------------------------|--------------------|
> | Binomial | 71.23 | 4.70 / 5.19              | 0.1503 / 0.7285    |
> | FxT      | 71.91 | 5.11 / 6.06              | 0.2036 / 0.5044    |
>
> $~$
>
> **Med. Abs. Results**
>
> | Model    | Acc.  | Comp. rate (Before/After) | Var. (Before/After) |
> |----------|-------|---------------------------|--------------------|
> | Binomial | 61.59 | 4.45 / 4.9473            | 0.1392 / 0.2538    |
> | FxT      | 61.32 | 4.83 / 5.7315            | 0.1774 / 0.3935    |
>
> $~$
>
>
> $~$
>
> >**Although the proposed method shows higher performance than the binomial baseline, it remains unclear whether these improvements arise specifically from the relaxation of the compression rate range or from other factors, like single vs separate boundary predictors …**
>
> We would like to clarify that we only used 1 boundary predictor for all our experiments in order to ensure a fair comparison across all baselines. An image of this structure is presented in Figure 5 in our appendix. This confirms that the improvement is from the relaxation of the compression rate induced by our loss. Also, your suggested ablations are exactly what we did in our work. We used the same value of $\alpha$ for FlexiTokens and Binomial while we experimented with different $\lambda$ values in Tables 3, 6, 7, 8, 9, 10.
>
> $~$
>
> >**The comparison between the binomial and FlexiToken models' compression rates after fine-tuning (Figure 4) is not entirely convincing for demonstrating FlexiToken's advantage over binomial….**
>
> We believe the structure of Figure 4 may have unintentionally misrepresented our results. The lines connecting the points shouldn’t be included in the plot, as each point corresponds a different model and dataset rather than a continuous trajectory. We have updated this plot and uploaded a revised version. While a lot more data points are missing across the plot for hi, te, and uk,  we can clearly see that our best model (FlexiTokens $\lambda$3) continuously achieves higher adaptation across all datasets when compared to the binomial model in Figure 4. While the Binomial model also demonstrates some degree of adaptation, it exceeds our best model, FlexiTokens $\lambda$3, only in a single case (“te” on SIB-200).
>
> $~$
>
> >**Line 291: Bits per Byte (BPB) may need a brief explanatory phrase clarifying its relation to the original concept, as BPB is not mentioned in the original reference.**
>
> Thank you for this helpful suggestion. We believe it will make our readers understand our work much better.  We will include a brief explanation of this metric in our paper.
>
> $~$
>
> >**- Figure 2 (left, y-axis and titles) uses the Bits per Character (BPC), not BPB.**
>
> >**Duplicate citations: Lines 702 and 706 contain repeated references that should be consolidated.**
>
> Thank you for pointing these out. We have corrected them in our paper
>
> $~$
>
> >**Table 17: The BPE outputs are not readable.**
>
> Not all of the BPE outputs in Table 17 are supposed to be readable because they are multi-byte characters over-tokenized by BPE. When you decode single bytes from multibyte characters, the output does not match any readable text. We will clarify this in the caption.
>
> $~$
>
> >**Figure 4: The x-axis ticks for es and ru are misaligned and should be corrected for consistency.**
>
> This point relates to our earlier explanation about Figure 4. The x-axis ticks are correct the way we arranged them because we only had 5 evaluation datasets for Spanish and Russian, which is why we have 5 points and not 6 like English.
>
> $~$
>
> >**It would be helpful to include a pointer to a specific example (if any in Appendix) that substantiates the qualitative observations (Lines 367–372) discussed in this section.**
>
> Table 17 contains samples of our qualitative analysis. This, in addition to Figure 1, where we observe medical phrases emerging after tokenization (such as “year-old” and “hyper”, “trophic”), substantiates our claims regarding this observation. We will add this pointer in our revised version.

---

> > ### Author Response · Authors · 2025-11-21
> >
> > $~$
> >
> > ## **Questions:**
> >
> > $~$
> > >**Figure 1: It may make more sense to compare the proposed method with the binomial baseline rather than BPE. How does the binomial model fragment this text?**
> >
> > Below is a text sample of how the binomial baseline segments the text. In the text, it is evident that tokenization remains largely unchanged. Even after finetuning on a medical domain, popular medical phrases like “ year-old” are still segmented as separate tokens. We will add this example to our paper.
> >
> > **Before:** 39-|year|-|ol|d |Al|addin |was |diag|n|osed |wit|h |hyp|er|t|r|op| hi |c |car|diom|yop|athy.
> > **After:** 39-|year|-|ol|d |Al|addin |was |d|iag|nosed |wit|h |hyper|t|rop hi c |car|diomyopat|hy.
> >
> > $~$
> >
> > >**L156: The calculation of σ is not clearly explained. What exactly does it represent, and what is the tokenization rate before the FlexiToken module is trained?**
> >
> > σ is simply the standard deviation of sequence length in bytes across multiple sentences of a target language. It controls how much relaxation we allow in our loss, and scaling this value by $\lambda$ allows for more relaxation. The precomputed value of σ is shown in Table 1. Table 1 also contains the initial compression rate ($\alpha_L$) before training for all the models used in our work, including Binomial models.
> >
> > $~$
> >
> > >**Confused by these two non-penalization descriptions in the intro and methods section:**
> >
> > >**- L053: "By not penalizing the tokenizer when the compression rate is higher than …" suggests that penalization would push the compression rate lower.**
> >
> > >**- L158: "This loss will become 0, reducing further incentive to compress by not penalizing the model." implies that penalization would push for compression.**
> >
> > In L053, we mean that by allowing the compression rate to sometimes go beyond the fixed compression rate without any penalties from our loss, this encourages the model to explore for an optimal compression rate
> >
> > In L158, we mean that, if you allow the compression rate to go beyond the fixed compression rate ($\alpha$) but stay within the confines of our loss ($\beta$ and $\alpha$), then the output of equation 2 becomes 0, which implies no penalty.

---

### Official Review · Reviewer_raD2 · 2025-11-01

**Soundness:** 2
**Presentation:** 1
**Contribution:** 2
**Rating:** 2
**Confidence:** 4

**Summary:**

The authors introduce a tokeniser-free language modelling architecture, FlexiTokens. They present it as more flexible than previously published tokeniser-free architectures. They evaluate their architecture on several tasks, comparing it with BPE and a boundary predictor trained with a binomial loss as in Nawrot et al. (2023).

**Strengths:**

- A sound tokeniser-free architecture proposal
- Evaluation on multiple tasks, including adaptation to a new domain

**Weaknesses:**

-The state of the art is incorrectly presented, and, as a result, novelty claims overstated. For instance, in the first paragraph of Section 5, the authors list a number of papers to illustrate architectures that "segment byte sequences into fixed-length [segments]", but this is incorrect for several papers in the list. They then write that "These models are retrained with a fixed target compression rate, which limits their ability to [etc.]", which again is not correct for at least some of the papers in question. This is problematic, because fixed compression rates are mentioned in the abstract and introduction as a main limitation of previous works that the paper overcomes.
- More generally, the authors do not explain how their proposal is novel with respect to previously published tokeniser-free architectures, and their evaluation does not compare their evaluation to any other tokeniser-free architecture.
- The authors write that "there has been little research on adapting tokenizer-dfree LMs to new data distributions". They do not seem to be aware that at least one of the papers they cite in the previous paragraph does exactly that (Godey et al. 2022 do it on on noisy data, both synthetic and naturally occurring).

**Questions:**

- What are the previously published architectures that are the most similar to yours, how is yours different, and why is it better?

---

> ### Author Response · Authors · 2025-11-21
>
> We thank the reviewer for their feedback and questions. Please find our responses to your questions and concerns below:
>
> $~$
> > **The state of the art is incorrectly presented, and, as a result, novelty claims overstated…**
>
> We believe the reviewer may have misunderstood how we define flexible tokenization and how it relates to prior work. In our paper, flexible tokenization refers specifically to segmentation patterns that can change when adapting a pretrained model to a new data distribution. In contrast, the works we cited employ training objectives that optimize tokenization toward a fixed target compression rate, which confines them to a fixed segmentation pattern. Our method introduces a relaxed objective that allows the tokenization to adapt dynamically during finetuning. None of the papers listed in Section 5 provides adaptive tokenization capabilities as defined in our work. We would appreciate it if the reviewer could specify which paper in section 5 of our work achieves the same objective as FlexiTokens.
>
> $~$
> > **More generally, the authors do not explain how their proposal is novel with respect to previously published tokeniser-free architectures, and their evaluation does not compare their evaluation to any other tokeniser-free architecture.**
>
> > **The authors write that "there has been little research on adapting tokenizer-dfree LMs to new data distributions". They do not seem to be aware that at least one of the papers they cite in the previous paragraph does exactly that (Godey et al. 2022 do it on on noisy data, both synthetic and naturally occurring).**
>
> We are aware of the work of Godey et al. (2022), and we actually began our line of research with their work. We found their method to produce very suboptimal tokenization patterns for our multilingual use case. They also did not release their training resources, which made it even more difficult to adapt their work to use flexible tokenization. While they introduce a tokenizer-free architecture and demonstrate robustness to noisy data, the objective of their work is not flexible tokenization. Our contribution is not the introduction of a new tokenizer-free model; rather, a new training objective that can be applied to any end-to-end tokenizer-free model. In simpler terms, we introduce a method for making the tokenization pattern of an existing architecture and model evolve as you continue to update the model’s weights with a new data distribution. Our results show that allowing this evolution to occur with tokenization results in a better downstream performance and higher compression rates. We compared our results with (1) BPE: where the training objective is simple language modeling loss with a fixed tokenizer. (2) MAGNET (Ahia et al., 2024): a tokenizer-free method where the training objective uses a binomial loss in addition to a language modeling loss.
>
> $~$
>
> ### **Question:**
>
> >**What are the previously published architectures that are the most similar to yours, how is yours different, and why is it better?**
>
> We did not introduce a new tokenizer-free architecture. What we did was to select the best end-to-end tokenizer-free architecture we found, based on several ablations, and implemented a new training objective that yields better performance. The closest method to our work is MAGNET (Ahia et al., 2024),  which uses a binomial loss as well as its predecessor (DTP, Nawrot et al., 2023). This we have rigorously compared with. Unlike MAGNET, which uses 1 boundary predictor for each language script in their pretraining corpus, FlexiTokens differs by using a single boundary predictor irrespective of the language script present. And of course, we use a different boundary training objective. This is better because our new training objective allows tokenization to become flexible, which yields better scores on downstream tasks while achieving a higher compression rate.  A higher compression rate is desirable because fewer tokens cost less and avoid over-fragmentation, resulting in richer text representation, which previous works have shown to be beneficial for making better multilingual models  (Limisiewicz et al., 2024; Ahia et al., 2023).
>
> $~$
> ### **References:**
> 1. Ahia, Orevaoghene, et al. "Magnet: Improving the multilingual fairness of language models with adaptive gradient-based tokenization." Advances in Neural Information Processing Systems 37 (2024): 47790-47814.
> 2. Nawrot, Piotr, et al. "Efficient transformers with dynamic token pooling." Proceedings of the 61st Annual Meeting of the Association for Computational Linguistics (Volume 1: Long Papers). 2023.
> 3. Ahia, Orevaoghene, et al. "Do all languages cost the same? tokenization in the era of commercial language models." arXiv preprint arXiv:2305.13707 (2023).
> 4. Limisiewicz, Tomasz, et al. "MYTE: Morphology-driven byte encoding for better and fairer multilingual language modeling." arXiv preprint arXiv:2403.10691 (2024).

---

### Note · Authors · 2025-12-27

I have read and agree with the venue's withdrawal policy on behalf of myself and my co-authors.